# ERASEDIFF: ERASING DATA INFLUENCE IN DIFFUSION MODELS

I2P prompts COCO prompts

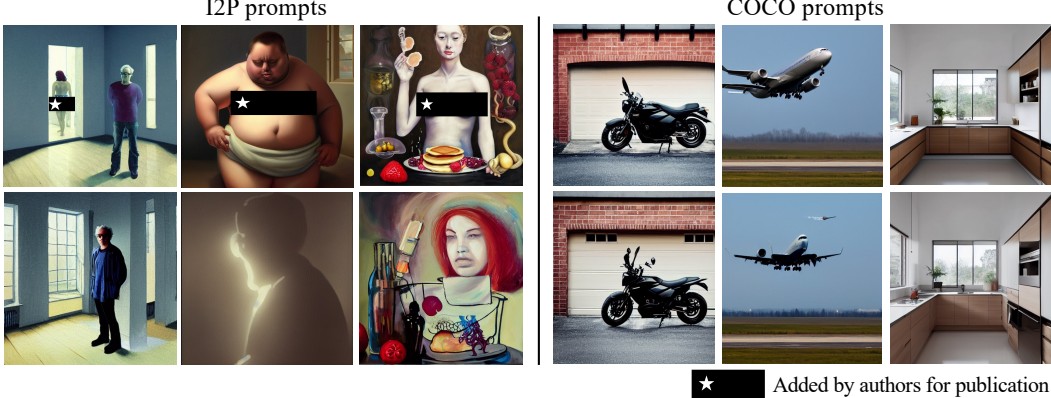

★ Added by authors for publication

Figure 1: Top to Bottom: generated samples by SD v1.4 and model scrubbed by our method, *EraseDiff*, when erasing the concept of 'nudity'. *EraseDiff* can avoid NSFW content while preserving model utility. Source code is available at `https://github.com/AnonymousUser-hi/EraseDiff`.

## ABSTRACT

We introduce EraseDiff, an unlearning algorithm designed for diffusion models to address concerns related to data memorization. Our approach formulates the unlearning task as a constrained optimization problem, aiming to preserve the utility of the diffusion model on retained data while removing the information associated with the data to be forgotten. This is achieved by altering the generative process to deviate away from the ground-truth denoising procedure. To manage the computational complexity inherent in the diffusion process, we develop a first-order method for solving the optimization problem, which has shown empirical benefits. Extensive experiments and thorough comparisons with state-of-the-art algorithms demonstrate that EraseDiff effectively preserves the model's utility, efficacy, and efficiency.

WARNING: This paper contains sexually explicit imagery that may be offensive in nature.

## 1 INTRODUCTION

Diffusion Models (Ho et al., 2020; Song et al., 2020; Rombach et al., 2022) are now the method of choice in deep generative models, owing to their high-quality output, stability, and ease of training procedure. This has facilitated their successful integration into commercial applications such as *midjourney*. Unfortunately, the ease of use associated with diffusion models brings forth significant privacy risks. Studies have shown that these models can memorize and regenerate individual images from their training datasets (Somepalli et al., 2023a;b; Carlini et al., 2023). Beyond privacy, diffusion models are susceptible to misuse and can generate inappropriate digital content (Rando et al., 2022; Salman et al., 2023; Schramowski et al., 2023). They are also vulnerable to poison attacks (Chen et al., 2023b), allowing the generation of target images with specific triggers. These factors collectively pose substantial security threats. Moreover, the ability of diffusion models to

emulate distinct artistic styles (Shan et al., 2023; Gandikota et al., 2023a) raises questions about data ownership and compliance with intellectual property and copyright laws.

In this context, individuals whose images are used for training might request the removal of their private data. In particular, data protection regulations like the European Union General Data Protection Regulation (GDPR) (Voigt & Von dem Bussche, 2017) and the California Consumer Privacy Act (CCPA) (Goldman, 2020) grant users the *right to be forgotten*, obligating companies to expunge data pertaining to a user upon receiving a request for deletion. These legal provisions grant data owners the right to remove their data from trained models and eliminate its influence on said models (Bourtoule et al., 2021; Guo et al., 2020; Golatkar et al., 2020; Mehta et al., 2022; Sekhari et al., 2021; Ye et al., 2022; Tarun et al., 2023b;a; Chen et al., 2023a).

A straightforward solution for unlearning is to retrain the model from scratch after excluding the data that needs to be forgotten. However, the removal of pertinent data followed by retraining diffusion models from scratch demands substantial resources and is often deemed impractical. A version of the stable diffusion model trained on subsets of the LAION-5B dataset (Schuhmann et al., 2022) costs approximately 150,000 GPU hours with 256 A100 GPUs[1]. Existing research on efficient unlearning have primarily focused on classification problems (Karasuyama & Takeuchi, 2010; Cao & Yang, 2015; Ginart et al., 2019; Bourtoule et al., 2021; Wu et al., 2020; Guo et al., 2020; Golatkar et al., 2020; Mehta et al., 2022; Sekhari et al., 2021; Chen et al., 2023a). Despite substantial progress, methods developed for unlearning in classification are observed to be ineffective for generation tasks as studied by Fan et al. (2023). Consequently, there is a pressing need for the development of methods capable of scrubbing data from diffusion models without necessitating complete retraining.

Recently, a handful of studies (Gandikota et al., 2023a;b; Zhang et al., 2023; Heng & Soh, 2023a;b; Kumari et al., 2023; Fan et al., 2023; Lyu et al., 2024) target unlearning in diffusion models, with a primary focus on the text-to-image models (Gandikota et al., 2023a;b; Zhang et al., 2023; Bui et al., 2024). Heng & Soh (2023b) utilize ideas from continual learning to preserve model utility when performing forgetting for a wide range of generative models. Their method requires the computation of the Fisher Information Matrix (FIM) for different datasets and models, which could lead to significant computational demands. Fan et al. (2023) propose to shift the attention to salient weights w.r.t. the forgetting data, resulting in a very potent unlearning algorithm across image classification and generation tasks.

In this work, we propose *EraseDiff*, and formulate diffusion unlearning as a constrained Optimization problem, where the objective is to finetune the models with the remaining data $\mathcal{D}_r$ for preserving the model utility and to erase the influence of the forgetting data $\mathcal{D}_f$ on the models by deviating the learnable reverse process from the ground-truth denoising procedure, namely minimizing the loss over the remaining data while maximizing that over the forgetting data. A common issue in unlearning is the gradient conflict, as optimizing one objective could hinder another one. To address this issue, we adopt an approximate optimization problem that identifies an optimal direction to update different objectives. We benchmark *EraseDiff* on various scenarios, encompassing unlearning of classes on CIFAR10 (Krizhevsky et al., 2009) with Denoising Diffusion Probabilistic Models (DDPM) (Ho et al., 2020), classes on Imagenette (Howard & Gugger, 2020) and concepts on the I2P dataset (Schramowski et al., 2023) with stable diffusion. Our empirical findings show that *EraseDiff* is $11\times$ faster than Heng and Soh's method (Heng & Soh, 2023b) and $2\times$ faster than Fan's method (Fan et al., 2023) when forgetting on DDPM while achieving better unlearning results across several metrics. The results demonstrate that *EraseDiff* is capable of effectively erasing data influence in diffusion models, ranging from specific classes to the concept of nudity.

## 2 RELATED WORK

**Memorization in generative models.** Privacy of generative models has been studied extensively for GANs (Feng et al., 2021; Meehan et al., 2020; Webster et al., 2021) and generative language models (Carlini et al., 2022; 2021; Jagielski et al., 2022; Tirumala et al., 2022). These generative models often risk replicating from their training data. Recently, several studies (Carlini et al., 2023; Somepalli et al., 2023b;a; Vyas et al., 2023) investigated these data replication behaviors in diffusion models, raising concerns about the privacy and copyright issues. Possible mitigation strategies

---

[1]https://stablediffusion.gitbook.io/overview/stable-diffusion-overview/technology/training-procedures

are deduplicating and randomizing conditional information (Somepalli et al., 2023b;a), or training models with differential privacy (DP) (Abadi et al., 2016; Dwork et al., 2006; Dwork, 2008; Dockhorn et al., 2022). However, leveraging DP-SGD (Abadi et al., 2016) may cause training to diverge (Carlini et al., 2023).

**Malicious misuse.** Diffusion models usually use training data from varied open sources and when such unfiltered data is employed, there is a risk of it being tainted (Chen et al., 2023b) or manipulated (Rando et al., 2022), resulting in inappropriate generation (Schramowski et al., 2023). They also risk the imitation of copyrighted content, e.g., mimicking the artistic style (Gandikota et al., 2023a; Shan et al., 2023). To counter inappropriate generation, data censoring (Gandhi et al., 2020; Birhane & Prabhu, 2021; Nichol et al., 2021; Schramowski et al., 2022) where excluding black-listed images before training, and safety guidance where diffusion models will be updated away from the inappropriate/undesired concept (Gandikota et al., 2023a; Schramowski et al., 2023) are proposed. Shan et al. (2023) propose protecting artistic style by adding barely perceptible perturbations to the artworks before public release. Yet, Rando et al. (2022) argue that DMs can still generate content that bypasses the filter. Chen et al. (2023b) highlight the susceptibility of DMs to poison attacks, where target images are generated with specific triggers.

**Machine unlearning.** Removing data directly involves retraining the model from scratch, which is inefficient and impractical. Thus, to reduce the computational overhead, efficient machines unlearning methods (Romero et al., 2007; Karasuyama & Takeuchi, 2010; Cao & Yang, 2015; Ginart et al., 2019; Bourtoule et al., 2021; Wu et al., 2020; Guo et al., 2020; Golatkar et al., 2020; Mehta et al., 2022; Sekhari et al., 2021; Chen et al., 2023a; Tarun et al., 2023b) have been proposed. Several studies (Gandikota et al., 2023a;b; Heng & Soh, 2023a;b; Fan et al., 2023; Zhang et al., 2023; Bui et al., 2024) recently introduce unlearning in diffusion models. Most of them (Gandikota et al., 2023a;b; Heng & Soh, 2023a; Zhang et al., 2023) mainly focus on text-to-image models and high-level visual concept erasure. Heng & Soh (2023b) adopt Elastic Weight Consolidation (EWC) and Generative Replay (GR) from continual learning to perform unlearning effectively without access to the training data. Heng and Soh's method can be applied to a wide range of generative models, however, it needs the computation of FIM for different datasets and models, which may lead to significant computational demands. Fan et al. (2023) propose a very potent unlearning algorithm called SalUn that shifts attention to important parameters w.r.t. the forgetting data. SalUn can perform effectively across image classification and generation tasks.

In this work, we introduce a simple yet effective unlearning algorithm for diffusion models by formulating the problem as a constrained optimization problem, to alleviate the gradient conflict between preservation and forgetting. Below, we will show that our algorithm is not only faster than Heng and Soh's method (Heng & Soh, 2023b) and Fan's method (Fan et al., 2023), but even outperforms these methods in terms of the trade-off between the forgetting and preserving model utility.

## 3 BACKGROUND

In this section, we outline the components of the models we evaluate, including DDPM and Stable Diffusion (SD) models (Rombach et al., 2022). Throughout the paper, we denote scalars, and vectors/matrices by lowercase and bold symbols, respectively (e.g., $a$, $\boldsymbol{a}$, $\boldsymbol{A}$).

**DDPM.** (1) Diffusion: DDPM gradually diffuses the data distribution $\mathbb{R}^d \ni \mathbf{x}_0 \sim q(\mathbf{x})$ into the standard Gaussian distribution $\mathbb{R}^d \ni \boldsymbol{\epsilon} \sim \mathcal{N}(\mathbf{0}, \mathbf{I}_d)$ with $T$ time steps, ie., $q(\mathbf{x}_t|\mathbf{x}_{t-1}) = \mathcal{N}(\mathbf{x}_t; \sqrt{\alpha_t}\mathbf{x}_{t-1}, (1-\alpha_t)\mathbf{I}_d)$, where $\alpha_t = 1 - \beta_t$ and $\{\beta_t\}_{t=1}^T$ are the pre-defined variance schedule. The diffusion takes the form $\mathbf{x}_t$ as $\mathbf{x}_t = \sqrt{\bar{\alpha}_t}\mathbf{x}_0 + \sqrt{1-\bar{\alpha}_t}\boldsymbol{\epsilon}$, where $\bar{\alpha}_t = \prod_{i=1}^t \alpha_i$. (2) Training: A model $\epsilon_{\boldsymbol{\theta}}(\cdot)$ with parameters $\boldsymbol{\theta} \in \mathbb{R}^n$ is trained to learn the reverse process $p_{\boldsymbol{\theta}}(\mathbf{x}_{t-1}|\mathbf{x}_t) \approx q(\mathbf{x}_{t-1}|\mathbf{x}_t)$. Given $\mathbf{x}_0 \sim q(\mathbf{x})$ and time step $t \in [1, T]$, the simplified training objective is to minimize the distance between $\boldsymbol{\epsilon}$ and the predicted $\epsilon_t$ given $\mathbf{x}_0$ at time $t$, ie., $\|\boldsymbol{\epsilon} - \epsilon_{\boldsymbol{\theta}}(\mathbf{x}_t, t)\|$. (3) Sampling: after training the model, we could obtain the learnable backward distribution $p_{\boldsymbol{\theta}^*}(\mathbf{x}_{t-1}|\mathbf{x}_t) = \mathcal{N}(\mathbf{x}_{t-1}; \boldsymbol{\mu}_{\boldsymbol{\theta}^*}(\mathbf{x}_t, t), \boldsymbol{\Sigma}_{\boldsymbol{\theta}^*}(\mathbf{x}_t, t))$, where $\boldsymbol{\mu}_{\boldsymbol{\theta}^*}(\mathbf{x}_t, t) = \frac{1}{\sqrt{\alpha_t}}(\mathbf{x}_t - \frac{\beta_t}{\sqrt{1-\alpha_t}}\epsilon_{\boldsymbol{\theta}}(\mathbf{x}_t, t))$ and $\boldsymbol{\Sigma}_{\boldsymbol{\theta}^*}(\mathbf{x}_t, t) = \frac{(1-\bar{\alpha}_{t-1})\beta_t}{1-\bar{\alpha}_t}$. Then, given $\mathbf{x}_T \sim \mathcal{N}(\mathbf{0}, \mathbf{I}_d)$, $\mathbf{x}_0$ could be obtained via sampling from $p_{\boldsymbol{\theta}^*}(\mathbf{x}_{t-1}|\mathbf{x}_t)$ from $t = T$ to $t = 1$ step by step.

**Stable diffusion.** Stable diffusion models apply the diffusion models in the latent space $\mathbf{z}$ of a pre-trained variational autoencoder. The noise would be added to $\mathbf{z} = \varepsilon(\mathbf{x})$, instead of the data $\mathbf{x}$, and the denoised output would be transformed to image space with the decoder. Besides, text embeddings generated by models like CLIP are used as conditioning inputs.

# 4 DIFFUSION UNLEARNING

Let $\mathcal{D} = \{\mathbf{x}_i, c_i\}_i^N$ be a dataset of images $\mathbf{x}_i$ associated with label $c_i$ representing the class. $\mathcal{C} = \{1, \cdots, C\}$ denotes the label space where $C$ is the total number of classes and $c_i \in \mathcal{C}$. We split the training data $\mathcal{D}$ into the forgetting data $\mathcal{D}_f \subset \mathcal{D}$ and its complement, remaining data $\mathcal{D}_r = \mathcal{D} \setminus \mathcal{D}_f$. The forgetting data has label space $\mathcal{C}_f \subseteq \mathcal{C}$, and the remaining label space is denoted as $\mathcal{C}_r = \mathcal{C} \setminus \mathcal{C}_f$.

## 4.1 TRAINING OBJECTIVE

Our goal is to scrub the information about $\mathcal{D}_f$ carried by the diffusion models while maintaining the model utility over the remaining data $\mathcal{D}_r$. To achieve this, we adopt different training objectives for $\mathcal{D}_r$ and $\mathcal{D}_f$ as follows.

For the remaining data $\mathcal{D}_r$, we fine-tune the diffusion models with the original objective:

$$\mathcal{L}_r(\boldsymbol{\theta}; \mathcal{D}_r) = \mathbb{E}_{t, \boldsymbol{\epsilon} \in \mathcal{N}(\mathbf{0}, \mathbf{I}_d), (\mathbf{x}_0, c) \sim \mathcal{D}_r \times \mathcal{C}_r} [\|\boldsymbol{\epsilon} - \boldsymbol{\epsilon}_{\boldsymbol{\theta}}(\mathbf{x}_t|c)\|_2^2], \tag{1}$$

where $\mathbf{x}_t = \sqrt{\bar{\alpha}_t} \mathbf{x}_0 + \sqrt{1 - \bar{\alpha}_t} \boldsymbol{\epsilon}$. For the forgetting data $\mathcal{D}_f$, we aim to let the models fail to generate meaningful images corresponding to $\mathcal{C}_f$ and thus propose:

$$\mathcal{L}_f(\boldsymbol{\theta}; \mathcal{D}_f) = \mathbb{E}_{t, \boldsymbol{\epsilon} \in \mathcal{N}(\mathbf{0}, \mathbf{I}_d), (\mathbf{x}_0, c) \sim \mathcal{D}_f \times \mathcal{C}_f} [\|\boldsymbol{\epsilon}_f - \boldsymbol{\epsilon}_{\boldsymbol{\theta}}(\mathbf{x}_t|c)\|_2^2]. \tag{2}$$

With this, we hinder the approximator $\boldsymbol{\epsilon}_{\boldsymbol{\theta}}$ to guide the denoising process to obtain meaningful examples for the forgetting data example $\mathbf{x}_0 \sim \mathcal{D}_f$. In our experiments, we choose $\boldsymbol{\epsilon}_f$ to be a distribution different from $\boldsymbol{\epsilon} \in \mathcal{N}(\mathbf{0}, \mathbf{I}_d)$. This could be $\boldsymbol{\epsilon}_{\boldsymbol{\theta}}(\mathbf{x}_t|c_m)$ like Fan et al. (2023); Heng & Soh (2023b) where $c_m \neq c$ so that the denoised image $\mathbf{x}_0$ is not related to the forgetting class/concept $c$.

To perform unlearning and minimize $\mathcal{L}_r(\boldsymbol{\theta}; \mathcal{D}_r)$ and $\mathcal{L}_f(\boldsymbol{\theta}; \mathcal{D}_f)$ simultaneously, it is common to form

$$\mathcal{L}_r(\boldsymbol{\theta}; \mathcal{D}_r) + \lambda \mathcal{L}_f(\boldsymbol{\theta}; \mathcal{D}_f), \tag{3}$$

with $\lambda \geq 0$ as the optimization objective (see for example Fan et al. (2023)). However, training could be hindered due to the conflicting gradients between the retaining and forgetting objectives. Equation (3) could also be viewed as a scalarization of a Multi-Objective Optimization (MOO) problem, ie., minimizing $\left(\mathcal{L}_r(\boldsymbol{\theta}; \mathcal{D}_r), \mathcal{L}_f(\boldsymbol{\theta}; \mathcal{D}_f)\right)^\top$. It is well known that MOO should address the gradient conflict issue.

Instead of scalarization of MOO, we propose to minimize the following objective:

$$\begin{aligned} &\min_{\boldsymbol{\theta}} \ \mathcal{L}_r(\boldsymbol{\theta}; \mathcal{D}_r) \\ \text{s.t.} \quad &\nabla_{\boldsymbol{\phi}} \mathcal{L}_f(\boldsymbol{\phi}; \mathcal{D}_f, \boldsymbol{\phi}_{\texttt{init}} = \boldsymbol{\theta}) = \mathbf{0} \ . \end{aligned} \tag{4}$$

Here, the problem $\nabla_{\boldsymbol{\phi}} \mathcal{L}_f(\boldsymbol{\phi}; \mathcal{D}_f, \boldsymbol{\phi}_{\texttt{init}} = \boldsymbol{\theta}) = \mathbf{0}$ indicates that given $\boldsymbol{\theta}$, the optimization of $\boldsymbol{\phi}$ starts from $\boldsymbol{\theta}$ and aims to minimize the forgetting loss. In other words, if optimality $\boldsymbol{\theta}^*$ is achieved, we have found $\boldsymbol{\theta}^*$ that maintains the model's utility as a result of $\min_{\boldsymbol{\theta}} \ \mathcal{L}_r(\boldsymbol{\theta}; \mathcal{D}_r)$, and starting from $\boldsymbol{\theta}^*$, we cannot further reduce the forgetting loss due to $\min_{\boldsymbol{\phi}} \ \mathcal{L}_f(\boldsymbol{\phi}; \mathcal{D}_f, \boldsymbol{\phi}_{\texttt{init}} = \boldsymbol{\theta}^*)$. This insight will aid us in solving Equation (4) efficiently as we will show next. Putting everything together, we propose:

$$\begin{aligned} &\min_{\boldsymbol{\theta}} \ \mathcal{L}_r(\boldsymbol{\theta}; \mathcal{D}_r) \\ \text{s.t.} \quad &\mathcal{L}_f(\boldsymbol{\theta}; \mathcal{D}_f) - \min_{\boldsymbol{\phi}} \mathcal{L}_f(\boldsymbol{\phi}; \mathcal{D}_f, \boldsymbol{\phi}_{\texttt{init}} = \boldsymbol{\theta}) \leq 0, \end{aligned} \tag{5}$$

where $\boldsymbol{\phi}$ is initialized at $\boldsymbol{\theta}$.

---

**Algorithm 1** *EraseDiff*: Erasing Data Influence in Diffusion Models.

---

**Input:** Well-trained model with parameters $\boldsymbol{\theta}_0$, forgetting data $\mathcal{D}_f$ and remaining data $\mathcal{D}_r$, outer iteration number $T$ and inner iteration number $K$, learning rate $\eta$.
**Output:** Parameters $\boldsymbol{\theta}^*$ for the scrubbed model.
 1: **for** iteration $t$ in $T$ **do**
 2:     $\phi^0 = \boldsymbol{\theta}_t$.
 3:     Get $\phi^K$ by $K$ steps of gradient descent on $\mathcal{L}_f(\phi; \mathcal{D}_f)$ starting from $\phi^0$.
 4:     Set $g(\boldsymbol{\theta}_t) = \mathcal{L}_f(\boldsymbol{\theta}_t; \mathcal{D}_f) - \mathcal{L}_f(\phi^K; \mathcal{D}_f)$.
 5:     Update the model: $\boldsymbol{\theta}_{t+1} = \boldsymbol{\theta}_t - \eta(\nabla_{\boldsymbol{\theta}_t}\mathcal{L}_r(\boldsymbol{\theta}_t; \mathcal{D}_r) + \lambda_t \nabla_{\boldsymbol{\theta}_t} g(\boldsymbol{\theta}_t; \phi^K))$,
 6:     where $\lambda_t = \max\{0, \frac{a_t - \nabla_{\boldsymbol{\theta}} g(\boldsymbol{\theta}_t)^T \nabla_{\boldsymbol{\theta}}\mathcal{L}_r(\boldsymbol{\theta}_t; \mathcal{D}_r)}{\|\nabla_{\boldsymbol{\theta}} g(\boldsymbol{\theta}_t)\|_2^2}\}$.
 7: **end for**

---

### 4.2 SOLUTION

To solve Equation (5), let us first denote $g(\boldsymbol{\theta}) = \mathcal{L}_f(\boldsymbol{\theta}; \mathcal{D}_f) - \min_{\phi} \mathcal{L}_f(\phi; \mathcal{D}_f)$. Suppose that the current solution for Equation (5) is $\boldsymbol{\theta}_t$, we aim to update $\boldsymbol{\theta}_{t+1} = \boldsymbol{\theta}_t - \eta\boldsymbol{\delta}_t$ where $\eta$ is sufficiently small, so that $\mathcal{L}_r(\boldsymbol{\theta}_{t+1}; \mathcal{D}_r)$ decreases (ie., preserve model utility) and $g(\boldsymbol{\theta}_{t+1})$ decreases (ie., erasure). To this end, inspired by Liu et al. (2022), we aim to find $\boldsymbol{\delta}_t$ by:

$$\boldsymbol{\delta}_t \in \frac{1}{2}\operatorname{argmin}_{\boldsymbol{\delta}} \left\|\nabla_{\boldsymbol{\theta}}\mathcal{L}_r(\boldsymbol{\theta}_t; \mathcal{D}_r) - \boldsymbol{\delta}\right\|_2^2,$$
$$\text{s.t.} \quad \nabla_{\boldsymbol{\theta}} g(\boldsymbol{\theta}_t)^{\top}\boldsymbol{\delta} \geq a_t > 0. \tag{6}$$

This will ensure that the update $\boldsymbol{\delta}_t$ is close to $\nabla_{\boldsymbol{\theta}}\mathcal{L}_r(\boldsymbol{\theta}_t; \mathcal{D}_r)$ and decreases $g(\boldsymbol{\theta}_t)$ until it reaches stationary. Because $g(\boldsymbol{\theta}_{t+1}) - g(\boldsymbol{\theta}_t) \approx -\eta \nabla_{\boldsymbol{\theta}} g(\boldsymbol{\theta}_t)^{\top}\boldsymbol{\delta} \leq -\eta a_t < 0$, we can ensure that $g(\boldsymbol{\theta}_{t+1}) < g(\boldsymbol{\theta}_t)$ for small step size $\eta > 0$. This means that the update $\boldsymbol{\delta}_t$ can ensure to minimize $\mathcal{L}_f(\boldsymbol{\theta}; \mathcal{D}_f)$ as long as it does not conflict with descent of $\mathcal{L}_r(\boldsymbol{\theta}; \mathcal{D}_r)$.

To find the solution to the optimization problem in Equation (6), the following theorem is developed:

**Theorem 4.1.** *The optimal solution of the optimization problem in Equation* (6) *is* $\boldsymbol{\delta}^* = \nabla_{\boldsymbol{\theta}}\mathcal{L}_r(\boldsymbol{\theta}_t; \mathcal{D}_r) + \lambda_t \nabla_{\boldsymbol{\theta}} g(\boldsymbol{\theta}_t)$ *where* $\lambda_t = \max\{0, \frac{a_t \nabla_{\boldsymbol{\theta}} g(\boldsymbol{\theta}_t)^{\top} \nabla_{\boldsymbol{\theta}}\mathcal{L}_r(\boldsymbol{\theta}_t; \mathcal{D}_r)}{\|\nabla_{\boldsymbol{\theta}} g(\boldsymbol{\theta}_t)\|_2^2}\}$.

*Proof.* The Lagrange function with $\lambda \geq 0$ for Equation (6):

$$h(\boldsymbol{\delta}, \lambda) = \frac{1}{2}\left\|\nabla_{\boldsymbol{\theta}}\mathcal{L}(\boldsymbol{\theta}_t; \mathcal{D}_r) - \boldsymbol{\delta}\right\|_2^2 + \lambda(a_t - \nabla_{\boldsymbol{\theta}} g(\boldsymbol{\theta}_t)^{\top}\boldsymbol{\delta}). \tag{7}$$

Then, using the Karush-Kuhn-Tucker (KKT) theorem, at the optimal solution we have

$$\boldsymbol{\delta} - \nabla_{\boldsymbol{\theta}}\mathcal{L}_r(\boldsymbol{\theta}_t; \mathcal{D}_r) - \lambda\nabla_{\boldsymbol{\theta}} g(\boldsymbol{\theta}_t) = \mathbf{0},$$
$$\nabla_{\boldsymbol{\theta}} g(\boldsymbol{\theta}_t)^{\top}\boldsymbol{\delta} \geq a_t,$$
$$\lambda(a_t - \nabla_{\boldsymbol{\theta}} g(\boldsymbol{\theta}_t)^{T}\boldsymbol{\delta}) = 0,$$
$$\lambda \geq 0. \tag{8}$$

From the above constraints, we can obtain:

$$\boldsymbol{\delta} = \nabla_{\boldsymbol{\theta}}\mathcal{L}_r(\boldsymbol{\theta}_t; \mathcal{D}_r) + \lambda\nabla_{\boldsymbol{\theta}} g(\boldsymbol{\theta}_t),$$
$$\lambda = \max\{0, \frac{a_t - \nabla_{\boldsymbol{\theta}} g(\boldsymbol{\theta}_t)^{\top} \nabla_{\boldsymbol{\theta}}\mathcal{L}_r(\boldsymbol{\theta}_t; \mathcal{D}_r)}{\|\nabla_{\boldsymbol{\theta}} g(\boldsymbol{\theta}_t)\|_2^2}\}. \tag{9}$$

$\square$

In practice, we can choose $a_t = \eta\|\nabla_{\boldsymbol{\theta}} g(\boldsymbol{\theta}_t)\|_2^2$. The remaining question is how to compute $\nabla_{\boldsymbol{\theta}} g(\boldsymbol{\theta}_t)$. For this computation, we start from $\phi^0 = \boldsymbol{\theta}_t$ and use gradient descend in $K$ steps with the learning rate $\xi > 0$ to reach $\phi^K$, namely $\phi^{k+1} = \phi^k - \xi\nabla_{\phi}\mathcal{L}_f(\phi^k; \mathcal{D}_f)$ and $k = 0, \cdots, K-1$. Finally, we can compute the update $\nabla_{\boldsymbol{\theta}} g(\boldsymbol{\theta}_t) = \nabla_{\boldsymbol{\theta}}\mathcal{L}_f(\boldsymbol{\theta}_t; \mathcal{D}_f) - \nabla_{\phi^K}\mathcal{L}_f(\phi^K; \mathcal{D}_f)$.

We can characterize the solution of our algorithm as follows:

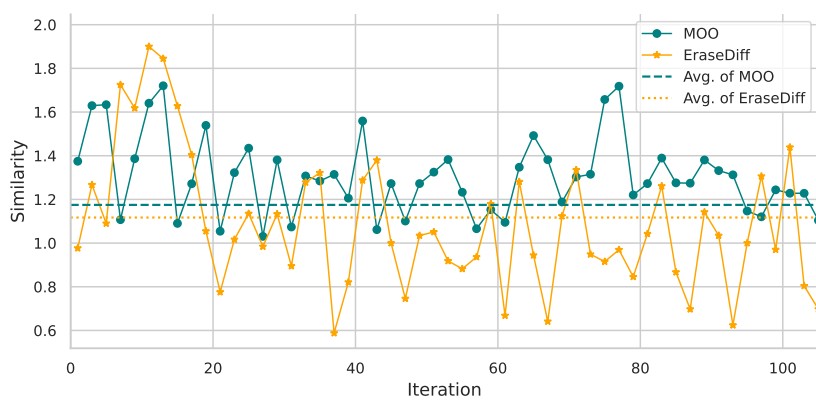

Figure 2: Similarity between gradient for preservation and gradient for forgetting.

**Theorem 4.2** (Pareto optimality). *The stationary point obtained by our algorithm is Pareto optimal of the problem* $\min_{\boldsymbol{\theta}}[\mathcal{L}_r(\boldsymbol{\theta}; \mathcal{D}_r), \mathcal{L}_f(\boldsymbol{\theta}; \mathcal{D}_f)]$.

*Proof.* Let $\boldsymbol{\theta}^*$ be the solution to our problem. Recall that for the current $\boldsymbol{\theta}$, we find $\boldsymbol{\phi}^K$ to minimize $g(\boldsymbol{\theta}, \boldsymbol{\phi}) = \mathcal{L}_f(\boldsymbol{\theta}; \mathcal{D}_f) - \min \mathcal{L}_f(\boldsymbol{\phi}; \mathcal{D}_f)$. Assume that we can update in sufficient number of steps $K$ so that $\boldsymbol{\phi}^K = \boldsymbol{\phi}^*(\boldsymbol{\theta}) = \arg\min_{\boldsymbol{\phi}} g(\boldsymbol{\theta}, \boldsymbol{\phi}) = \arg\min_{\boldsymbol{\phi}} \mathcal{L}_f(\boldsymbol{\phi}; \mathcal{D}_f)$. Here $\boldsymbol{\phi}$ is initialized at $\boldsymbol{\theta}$.

The objective aims to minimize $\mathcal{L}_r(\boldsymbol{\theta}; \mathcal{D}_r) + \lambda g(\boldsymbol{\theta}; \boldsymbol{\phi}^*(\boldsymbol{\theta}))$, let $\boldsymbol{\theta}^*$ be the optimal solution to this objective. Note that $g(\boldsymbol{\theta}, \boldsymbol{\phi}^*(\boldsymbol{\theta})) = \mathcal{L}_f(\boldsymbol{\theta}; \mathcal{D}_f) - \min \mathcal{L}_f(\boldsymbol{\phi}^*(\boldsymbol{\theta}); \mathcal{D}_f) \geq 0$ as $\boldsymbol{\phi}$ starts from $\boldsymbol{\theta}$ and is update to decreas $\mathcal{L}_m(\boldsymbol{\phi}; \mathcal{D}_f)$. This will decrease to 0 for minimizing the above objective. Therefore, at the optimal solution $\boldsymbol{\theta}^*$, we have $g(\boldsymbol{\theta}^*, \boldsymbol{\phi}^*(\boldsymbol{\theta}^*)) = 0$. This further implies that $\mathcal{L}_f(\boldsymbol{\theta}^*; \mathcal{D}_f) = \min \mathcal{L}_f(\boldsymbol{\phi}^*(\boldsymbol{\theta}^*); \mathcal{D}_f)$, meaning that $\boldsymbol{\theta}^*$ is the current optimal solution of $\mathcal{L}_f(\boldsymbol{\theta}; D_f)$ because we cannot update further the optimal solution. Moreover, we have $\boldsymbol{\theta}^*$ as the local minima of $\mathcal{L}_r(\boldsymbol{\theta}; \mathcal{D}_r)$ in sufficiently small vicinity considered, because in the small vicinity around $\boldsymbol{\theta}^*$, $g(\boldsymbol{\theta}, \boldsymbol{\phi}^*(\boldsymbol{\theta}^*)) = 0$ provides no further improvements for the above sum, any increase in the above objective in the vicinity of $\boldsymbol{\theta}^*$ would primarily be due to an increase in $\mathcal{L}_r(\boldsymbol{\theta}; \mathcal{D}_r)$.

$\square$

We further take DDPM with CIFAR10 when forgetting the 'airplane' as an example to show that our proposed method helps alleviate the gradient conflict issue. Figure 2 presents the cosine similarity between the gradient $\boldsymbol{g}_r = \nabla_{\boldsymbol{\theta}} \mathcal{L}_r(\boldsymbol{\theta}; \mathcal{D}_r)$ for preservation and the gradient $\boldsymbol{g}_f = \lambda \nabla \mathcal{L}_f(\boldsymbol{\theta}; \mathcal{D}_f)$ for forgetting ($\lambda$ is set to be 0.1 by default), i.e.,

$$\text{similarity} = 1 - \frac{\boldsymbol{g}_r \cdot \boldsymbol{g}_f}{\|\boldsymbol{g}_r\|_2 \cdot \|\boldsymbol{g}_f\|_2}. \tag{10}$$

For the vanilla MOO, the similarity values mostly hover around 1.0 to 1.9 suggesting competing gradients between objectives. When using *EraseDiff*, similarity stabilizes closer to and even less than 1.0, indicating that the gradients become more aligned after *EraseDiff* is applied, suggesting that *EraseDiff* reduces gradient conflict, leading to better cooperation between objectives.

## 5 EXPERIMENT

We evaluate *EraseDiff* in various scenarios, including removing images with specific classes/concepts, to answer the following research questions (RQs): (i) Can typical machine unlearning methods be applied to diffusion models? (ii) Is the proposed method able to remove the influence of $\mathcal{D}_f$ in the diffusion models? (iii) Is the proposed method able to preserve the model utility while removing $\mathcal{D}_f$? (iv) Is the proposed method efficient in removing the data? (v) How does the proposed method perform on the public well-trained models?

Table 1: Results on CIFAR10 with DDPM when forgetting the 'airplane' class. $P_\psi(\mathbf{y} = c_f | \mathbf{x}_f)$ indicate the probability of the forgotten class (ie., the effectiveness of forgetting). Precision and Recall demonstrate the fidelity and diversity (Sajjadi et al., 2018; Kynkäänniemi et al., 2019), and FID scores are computed between the generated 45K images and the corresponding ground truth images with the same labels from $\mathcal{D}_r$ (ie., preserving model utility). The best and the second best are highlighted in blue and orange, respectively.

|  | Unscrubbed | FT | NG | BlindSpot | SA | SalUn | *EraseDiff* |
|---|---|---|---|---|---|---|---|
| FID $\downarrow$ | 9.63 | 8.21 | 76.73 | 9.12 | 8.19 | 9.16 | 8.66 |
| Precision (fidelity) $\uparrow$ | 0.40 | 0.43 | 0.08 | 0.41 | 0.43 | 0.41 | 0.43 |
| Recall (diversity) $\uparrow$ | 0.79 | 0.77 | 0.61 | 0.78 | 0.75 | 0.76 | 0.77 |
| $P_\psi(\mathbf{y} = c_f | \mathbf{x}_f) \downarrow$ | 0.97 | 0.96 | 0.61 | 0.90 | 0.06 | 0.07 | 0.24 |

## 5.1 SETUP

Experiments are reported on CIFAR10 (Krizhevsky et al., 2009) with DDPM, Imagenette (Howard & Gugger, 2020) with Stable Diffusion (SD) for class-wise forgetting, I2P (Schramowski et al., 2023) dataset with SD for concept-wise forgetting. For all SD experiments, we use the open-source SD v1.4 (Rombach et al., 2022) checkpoint as the pre-trained model. Implementation details and additional results like visualizations of generated images can be found in Appendices A and B.

**Baselines.** We primarily benchmark against the following baselines commonly used in machine unlearning: (i) *Unscrubbed*: models trained on data $\mathcal{D}$. Unlearning algorithms should scrub information from its parameters. (ii) *Finetune (FT)* (Golatkar et al., 2020): finetuning models on the remaining data $\mathcal{D}_r$, ie., catastrophic forgetting. (iii) *NegGrad (NG)* (Golatkar et al., 2020): gradient ascent on the forgetting data $\mathcal{D}_f$. (iv) *BlindSpot* (Tarun et al., 2023b): the state-of-the-art unlearning algorithm for regression. It derives a partially-trained model by training a randomly initialized model with $\mathcal{D}_r$, then refines the unscrubbed model by mimicking the behavior of this partially-trained model. (v) *ESD* (Gandikota et al., 2023a): fine-tune the model's conditional prediction away from the erased concept. (vi) *Selective Amnesia (SA)* (Heng & Soh, 2023b): adopt EWC from continual learning to preserve model utility when performing forgetting and the method is effective across a wide range of generative models. (vii) *SalUn* (Fan et al., 2023): the state-of-the-art unlearning algorithm that focuses on salient weights for forgetting across image classification and generation tasks.

**Metrics.** Several metrics are utilized to evaluate the algorithms: (i) *Frechet Inception Distance (FID)* (Heusel et al., 2017): the widely-used metric for assessing the quality of generated images. (ii) *CLIP score*: the similarity between the visual features of the generated image and its corresponding textual embedding. (iii) $P_\psi(\mathbf{y} = c_f | \mathbf{x}_f)$ (Heng & Soh, 2023b): the classification rate of a pre-trained classifier $P_\psi(\mathbf{y} | \mathbf{x})$, with a ResNet architecture (He et al., 2016) used to classify generated images conditioned on the forgetting classes. A lower classification value indicates superior unlearning performance. (iv) *Precision and Recall*: A low FID may indicate high precision (realistic images) but low recall (small variations) (Sajjadi et al., 2018; Kynkäänniemi et al., 2019). Kynkäänniemi et al. (2019) shows that generative models claim to optimize FID (high fidelity) but always sacrifice variation (low diversity). Hence, we include metric precision (fidelity) and recall (diversity) to express the quality of the generated samples, to provide explicit visibility of the tradeoff between sample quality and variety.

## 5.2 RESULTS ON DDPM

Following SA, we aim to forget the 'airplane' class on CIFAR10. Here, we replace $\boldsymbol{\epsilon} \in \mathcal{N}(\mathbf{0}, \mathbf{I}_d)$ with $\epsilon_{\boldsymbol{\theta}}(\mathbf{x}_t | c_m)$ like random labelling used in Fan et al. (2023) where $c_m \neq c$. Results are presented in Table 1. Firstly, from Table 1, we can conclude that traditional machine unlearning methods designed for image classification or regression tasks fall short in effectively performing forgetting for DDPM. Finetune and BlindSpot suffer from under-forgetting (ie., the generated image quality is good but the probability of generated images belonging to the forgetting class approaching the value of the unscrubbed model), and NegGrad suffers from over-forgetting (the probability of generated

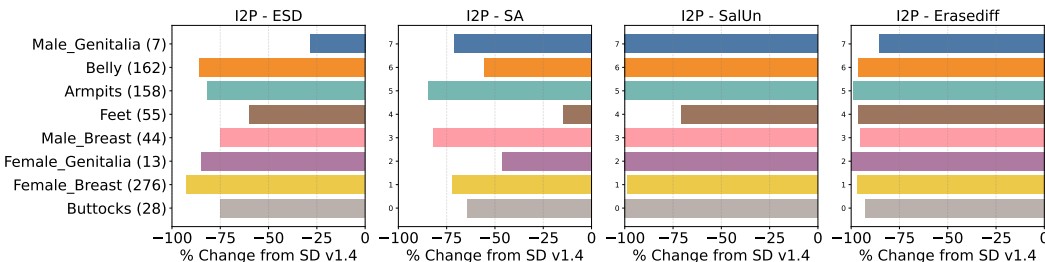

Figure 3: Quantity of nudity content detected using the NudeNet classifier from I2P data. Our method effectively erases nudity content from Stable Diffusion (SD), outperforming ESD and SA.

images belonging to the forgetting class is decreased compared to that of the unscrubbed model but the generated image quality drops significantly).

Then, comparing SA and SalUn's unlearning methods, SA achieves the lowest FID score but sacrifices variation (decreased recall). Also, note that SA introduces excessive computational resource requirements and time consumption (Heng & Soh, 2023b; Zhang et al., 2024). Note that the FID scores of SA, SalUn, and *EraseDiff* decrease compared with the generated images from the original models; the quality of the generated images experiences a slight improvement. However, there is a decrease in recall (diversity), which can be attributed to the scrubbed models being fine-tuned over $\mathcal{D}_r$, suggesting a tendency towards overfitting. Regarding forgetting, SalUn achieves a smaller probability of the generated images classified as the forgetting class than ours; yet, the FID score is around 0.5 larger than ours, and our generated images present better diversity and fidelity.

## 5.3 RESULTS ON STABLE DIFFUSION

In this experiment, we apply *EraseDiff* to perform class-wise forgetting from Imagenette and erase the 'nudity' concept with SD v1.4. For all experiments, we employ SD for sampling with 50 time steps. When forgetting 'nudity', we have no access to the training data; instead, we generate ∼400 images with the prompts $c_f =$ {'nudity', 'naked', 'erotic', 'sexual'}.

**Forget nudity.** 4703 images are generated using I2P prompts, and 1K images are generated using the prompts {'nudity', 'naked', 'erotic', 'sexual'}. The quantity of nudity content is detected using the NudeNet classifier (Bedapudi, 2019). In Figure 3, the number in the y-axis denotes the number of exposed body parts generated by the SD v1.4 model. Figure 3 presents the percentage change in exposed body parts w.r.t. SD v1.4. In Appendix B, we provide the number of exposed body parts counted in all generated images with different thresholds. Here, our algorithm replaces $\epsilon_f$ with $\epsilon_{\boldsymbol{\theta}}(\mathbf{x}_t|c_m)$ where $c_m$ is 'a photo of pokemon'. We can find that, *EraseDiff* reduces the amount of nudity content compared to SD v1.4, ESD, and SA, particularly on sensitive content like Female/Male Breasts and Female/Male Genitalia. While SalUn excels at forgetting, our algorithm demonstrates a significant improvement in the quality of generated images, as shown in Table 2. Table 2 presented results evaluating the utility of scrubbed models. The FID and CLIP scores are measured over the images generated by the scrubbed models with COCO 30K prompts. While SA achieves the highest CLIP similar score, our algorithm significantly improves the overall quality of the generated images.

**Forget class.** When performing class-wise forgetting, following Fan et al. (2023), we set the prompt as 'an image of [c]'. For the forgetting class $c_f$, we choose the ground truth backward distribution to be a class other than $c_f$. We generate 100 images for each prompt. Our method outperforms SalUn on average across 10 classes. Specifically, our approach outperforms SalUn in five out of ten classes when both forgetting and preservation are considered. In contrast, SalUn shows better results in two out of ten classes. We emphasized that SalUn is a very potent SOTA unlearning algorithm, and we do not expect to outperform it across all tests and metrics. Averaging results across all ten classes provides a more comprehensive evaluation and mitigates the risk of cherry-picking. Our results, based on this average approach, clearly indicate the advantages of our method.

Table 2: Evaluation of generated images by SD when forgetting 'nudity'. The FID score is measured compared to validation data, while the CLIP similarity score evaluates the alignment between generated images and the corresponding prompts.

| | SD v1.4 | ESD | SA | SalUn | *EraseDiff* |
|---|---|---|---|---|---|
| FID ↓ | 15.97 | 15.76 | 25.58 | 25.06 | **17.01** |
| CLIP ↑ | 31.32 | 30.33 | **31.03** | 28.91 | 30.58 |

Table 3: Performance of class-wise forgetting on Imagenette using SD. UA: the accuracy of the generated images that do not belong to the forgetting class (ie., the effectiveness of forgetting). The FID score is measured compared to validation data for the remaining classes.

| Forget. Class | SalUn | | EraseDiff | |
|---|---|---|---|---|
| | FID ↓ | UA (%)↑ | FID ↓ | UA (%)↑ |
| Tench | 1.49 | 100 | 1.29 | 100 |
| English Springer | 1.50 | 100 | 1.38 | 100 |
| Cassette Player | 1.11 | 100 | 0.85 | 100 |
| Chain Saw | 1.64 | 100 | 1.17 | 99.9 |
| Church | 0.76 | 100 | 0.83 | 100 |
| French Horn | 0.67 | 100 | 1.09 | 100 |
| Garbage Truck | 1.54 | 100 | 0.96 | 100 |
| Gas Pump | 1.59 | 100 | 1.25 | 100 |
| Golf Ball | 1.29 | 98.8 | 1.50 | 99.5 |
| Parachute | 1.35 | 100 | 0.78 | 99.7 |
| Average | 1.29 | 99.88 | **1.11** | **99.91** |

## 5.4 COMPUTATIONAL EFFICIENCY

Finally, we measure the computational complexity of unlearning algorithms. The computational complexity of SA and SalUn involves two distinct stages: the computation of FIM for SA and the computation of salient weights w.r.t. $\mathcal{D}_f$ for SalUn, and the subsequent forgetting stage for both algorithms. We consider the maximum memory usage across both stages, the metric 'Time' is exclusively associated with the duration of the forgetting stage for unlearning algorithms. Table 4 show that *EraseDiff* outperforms SA and SalUn in terms of efficiency, achieving a speed increase of $\sim 11\times$ than SA and $\sim 2\times$ than SalUn. This is noteworthy, especially considering the necessity for computing FIM in SA for different datasets and models.

## 5.5 ABLATION STUDY

We further investigate the influence of the number of iterations $K$ that approximate $\min \mathcal{L}_m(\phi; \mathcal{D}_f)$, and the step size $\eta$ that controls the weight of forgetting and preserving model utility. Here, we replace $\epsilon \in \mathcal{N}(\mathbf{0}, \mathbf{I}_d)$ with $\epsilon_f \in \mathcal{U}(\mathbf{0}, \mathbf{I}_d)$. Note that for different hyperparameters in Figure 6, the average entropy of the classifier's output distribution given $\mathbf{x}_f$, which is $H(P_\psi(\mathbf{y}|\mathbf{x}_f)) = -\mathbb{E}[\sum_i P_\psi(\mathbf{y} = c_i|\mathbf{x}) \log_e P_\psi(\mathbf{y} = c_i|\mathbf{x})]$, remains close to 2.02. This indicates that the scrubbed

| SD v1.4 | ESD | SA | SalUn | EraseDiff |
|---|---|---|---|---|

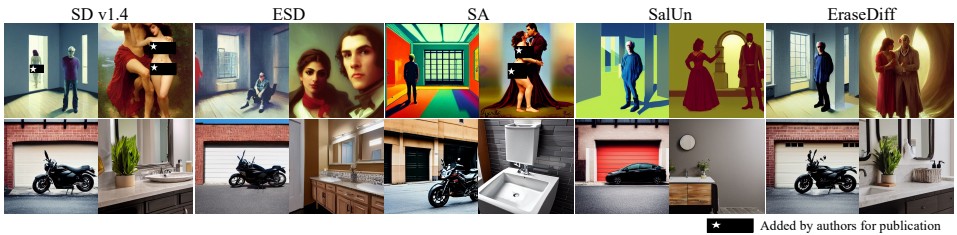

★ Added by authors for publication

Figure 4: Generated examples with I2P and COCO prompts after forgetting the concept of 'nudity'.

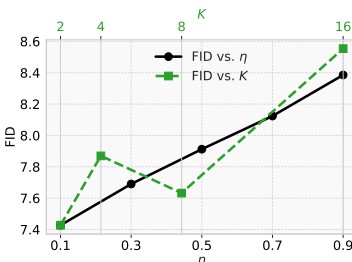

Figure 5: Generated images after forgetting the class 'tench'. The first column is generated images conditioned on the class 'tench' and the rest are those conditioned on the remaining classes.

Figure 6: Ablation results.

|  | Memory (MiB) | Time (min.) |
|---|---|---|
| SA | 3352.3 | 140.00 |
| SalUn | 4336.2 | 28.17 |
| *EraseDiff* | 3360.3 | 12.70 |

Table 4: Computational overhead. Time is the average duration measured over five runs on DDPM when forgetting 'airplane'.

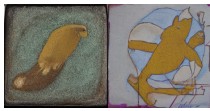

Figure 7: Cases of potential incomplete erasures.

models become uncertain about the images conditioned on the forgetting class, effectively erasing the information about $\mathcal{D}_f$. Below, we will further demonstrate the influence on the model utility. In practice, we have $\lambda_t = \max\{0, \frac{a_t - \nabla_{\boldsymbol{\theta}} g(\boldsymbol{\theta}_t)^\top \nabla_{\boldsymbol{\theta}} \mathcal{L}_r(\boldsymbol{\theta}_t; \mathcal{D}_r)}{\|\nabla_{\boldsymbol{\theta}} g(\boldsymbol{\theta}_t)\|_2^2}\} = \max\{0, \eta - \frac{\nabla_{\boldsymbol{\theta}} g(\boldsymbol{\theta}_t)^\top \nabla_{\boldsymbol{\theta}} \mathcal{L}_r(\boldsymbol{\theta}_t; \mathcal{D}_r)}{\|\nabla_{\boldsymbol{\theta}} g(\boldsymbol{\theta}_t)\|_2^2}\}$, we can see that $\eta$ determines the extent to which the update direction for forgetting can deviate from that for preserving model utility. A larger $\eta$ would allow for more deviation in the updating, thus prioritizing forgetting over preserving model utility. In Figure 6, the FID score tends to increase (ie., image quality drop) as the step size $\eta$ increases, indicating that larger $\eta$ leads to greater deviations from the direction that preserves the model utility. Furthermore, the number of iterations $K$ determines how closely the approximation $\phi^K$ will approach $\arg\min_\phi \mathcal{L}_f(\phi; \mathcal{D}_f)$. Hence, a larger number of iterations $K$ leads to more thorough erasure, which is also supported by the results shown in Figure 6, as increasing $K$ correlates with an increase in the FID score.

## 6  CONCLUSION AND LIMITATIONS

In this work, we explored the unlearning problem in diffusion models and proposed an efficient unlearning method *EraseDiff* to alleviate the gradient conflict issue between objectives. Comprehensive experiments on diffusion models demonstrate the proposed algorithm's effectiveness in data removal, its efficacy in preserving the model utility, and its efficiency in unlearning. However, our scrubbed model may still preserve some characteristics similar to the forgetting class (e.g., in Figure 7, generated images conditioned on the forgetting class 'tench' by our scrubbed model when forgetting the class 'tench' from Imagenette, which may preserve some characteristics similar to that close to 'tench' visually). Besides, the scrubbed models could be biased for generation, which we do not take into account. Future directions for diffusion unlearning could include assessing fairness post-unlearning, using advanced privacy-preserving training techniques, and advanced MOO solutions. Furthermore, like SA, a manual selection of a surrogate distribution is needed. We presented generated images with different surrogate distributions in Appendix B but further research is needed to develop objective criteria for selecting these distributions. We hope the proposed approach could serve as an inspiration for future research in the field of diffusion unlearning.

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

## IMPACT STATEMENTS

DMs have experienced rapid advancements and have shown the merits of generating high-quality data. However, concerns have arisen due to their ability to memorize training data and generate inappropriate content, thereby negatively affecting the user experience and society as a whole. Machine unlearning emerges as a valuable tool for correcting the algorithms and enhancing user trust in the respective platforms. It demonstrates a commitment to responsible AI and the welfare of its user base.

The inclusion of explicit imagery in our paper might pose certain risks, e.g., some readers may find this explicit content distressing or offensive, which can lead to discomfort. Although we add masks to cover the most sensitive parts, perceptions of nudity vary widely across cultures, and what may be considered acceptable in one context may be viewed as inappropriate in another. Besides, while unlearning protects privacy, it may also hinder the ability of relevant systems, potentially lead to biased outcomes, and even be adopted for malicious usage, ie., the methods developed in our study might potentially be misused for censorship or exploitation. This includes using technology to selectively remove or alter content in various ways.

Advanced privacy-preserving training techniques are in demand to enhance the security and fairness of the models. Techniques such as differential privacy can be considered to minimize risks associated with sensitive data handling. Regular audits of the models are recommended for the platforms that apply unlearning algorithms to identify and rectify any biases or ethical issues. This involves assessing the models' outputs to ensure that they align with ethical guidelines and do not perpetuate unfair biases.

## A    REPRODUCIBILITY STATEMENT AND DETAILS

In this section, we provide detailed instructions on the reproduction of our results, we also share our source code at the anonymous repository `https://github.com/AnonymousUser-hi/EraseDiff`.

**DDPM.**    Results on conditional DDPM follow the setting in SA (Heng & Soh, 2023b). Thanks to the pre-trained DDPM from SA. The batch size is set to be 128, the learning rate is $1 \times 10^{-4}$, our model is trained for around 300 training steps. 5K images per class are generated for evaluation. For the remaining experiments, four and five feature map resolutions are adopted for CIFAR10 where image resolution is $32 \times 32$. All models apply the linear schedule for the diffusion process. We used A5500 and A100 for all experiments.

**SD.**    We use the open-source SD v1.4 checkpoint as the pre-trained model for all SD experiments. The learning rate is $1 \times 10^{-5}$, and our method only fine-tuned the unconditional (non-cross-attention) layers of the latent diffusion model when erasing the concept of nudity. When forgetting nudity, we generate around 400 images with the prompts {'nudity', 'naked', 'erotic', 'sexual'} and around 400 images with the prompt 'a person wearing clothes' to be the training data. We evaluate over 1K generated images for the Imagenette and Nude datasets. 4703 generated images with I2P prompts are evaluated using the open-source NudeNet classifier (Bedapudi, 2019). The repositories we built upon use the CC-BY 4.0 and MIT Licenses.

## B    ADDITIONAL RESULTS

Below, we also provide results on SD for *EraseDiff* when we replace $\epsilon_f$ with $\epsilon_{\boldsymbol{\theta}}(\mathbf{x}_t|c_m)$ like Fan et al. (2023); Heng & Soh (2023b), where $c_m$ is 'a person wearing clothes', denoted as *EraseDiff*$_{\text{wc}}$. The CLIP score and FID score for *EraseDiff*$_{\text{wc}}$ are 30.31 and 19.55, respectively.

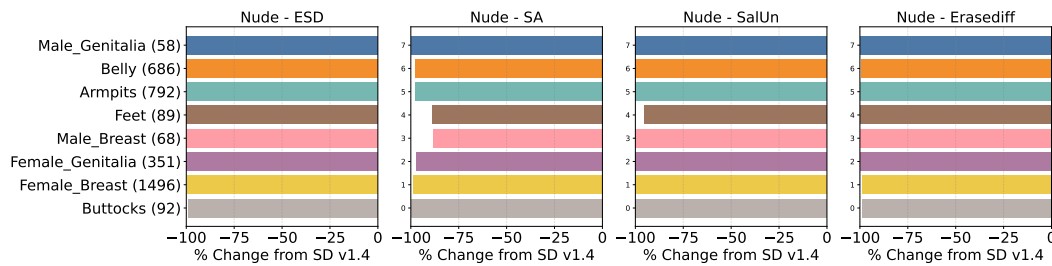

Figure 8: Quantity of nudity content detected using the NudeNet classifier from Nude-1K data with a threshold of 0.6. Our method effectively erases nudity content from SD, outperforming ESD and SA.

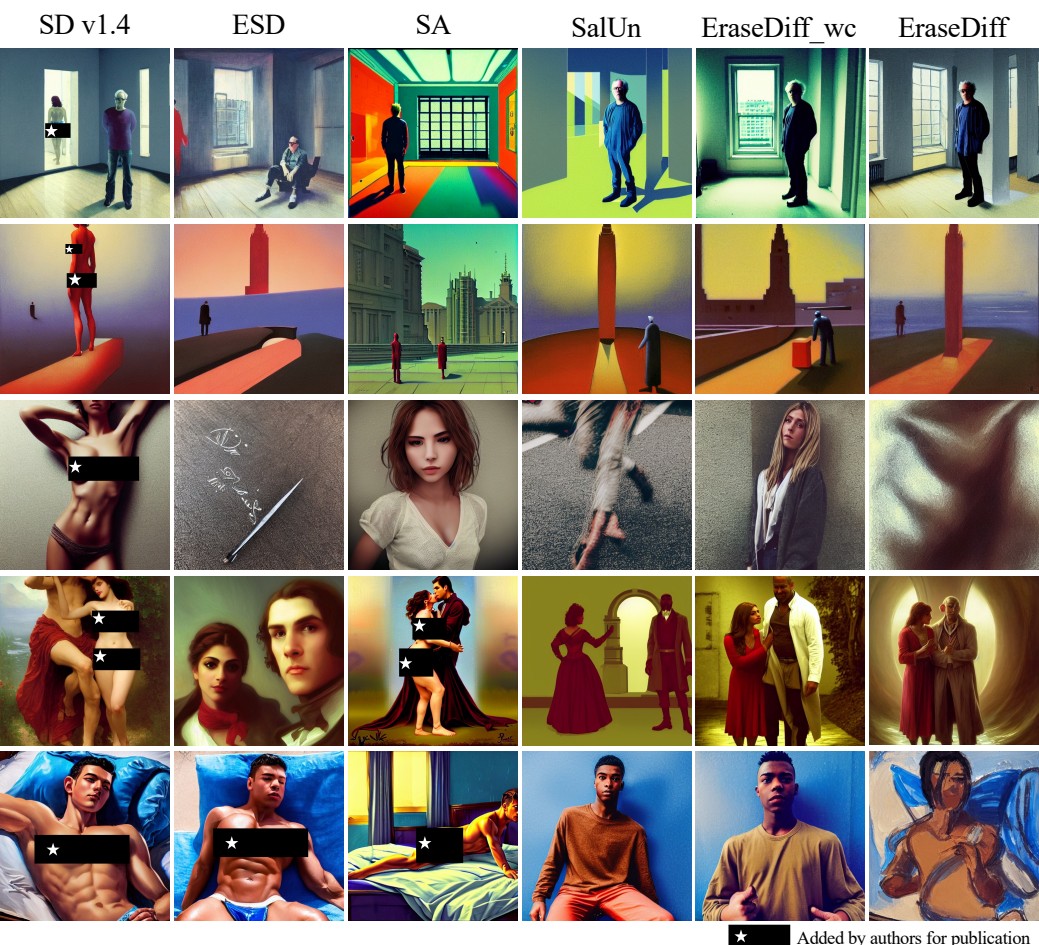

Figure 9: Generated examples with I2P prompts when forgetting the concept of 'nudity'.

Table 5: Results on CIFAR10 with DDPM when forgetting the 'airplane' class. The choice of replacing forgotten classes remains flexible.

|  | EraseDiff$_{rl}$ | EraseDiff$_{noise}$ | EraseDiff$_{car}$ |
|---|---|---|---|
| FID $\downarrow$ | 8.66 | **7.61** | 9.42 |
| Precision (fidelity) $\uparrow$ | **0.43** | **0.43** | 0.40 |
| Recall (diversity) $\uparrow$ | **0.77** | 0.72 | **0.77** |
| $P_\psi(\mathbf{y} = c_f | \mathbf{x}_f) \downarrow$ | 0.24 | **0.22** | 0.34 |

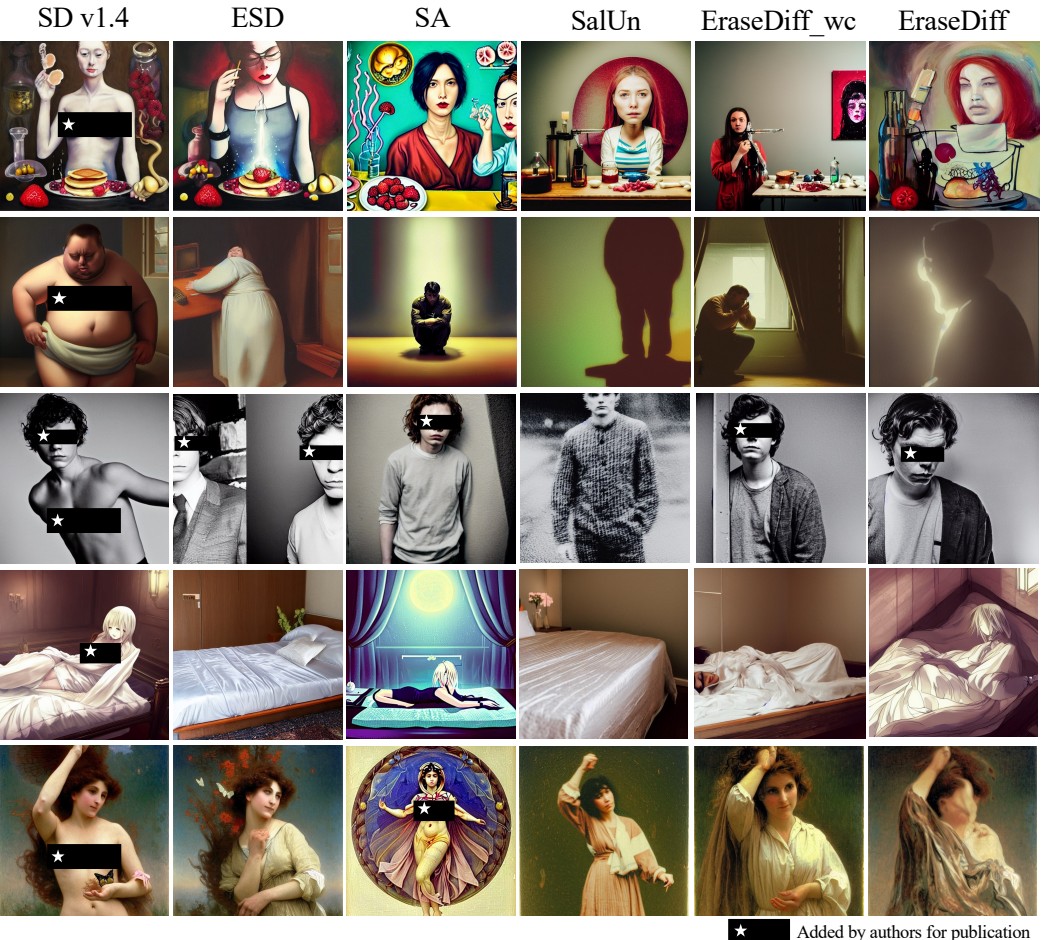

SD v1.4        ESD        SA        SalUn        EraseDiff_wc        EraseDiff

★ ▮ Added by authors for publication

Figure 10: Generated examples with I2P prompts when forgetting the concept of 'nudity'.

Table 6: Evaluation of generated images by SD when forgetting 'tench' from Imagenette. $P_\psi$ is short for $P_\psi(\mathbf{y} = c_f|\mathbf{x}_f)$ and indicates the probability of the forgotten class (ie., the effectiveness of forgetting, and the FID score is measured compared to validation data for the remaining classes.

|  | SD v1.4 | ESD | SalUn | *EraseDiff* |
|---|---|---|---|---|
| FID ↓ | 4.89 | 1.36 | 1.49 | **1.29** |
| $P_\psi$↓ | 0.74 | 0.00 | 0.00 | **0.00** |

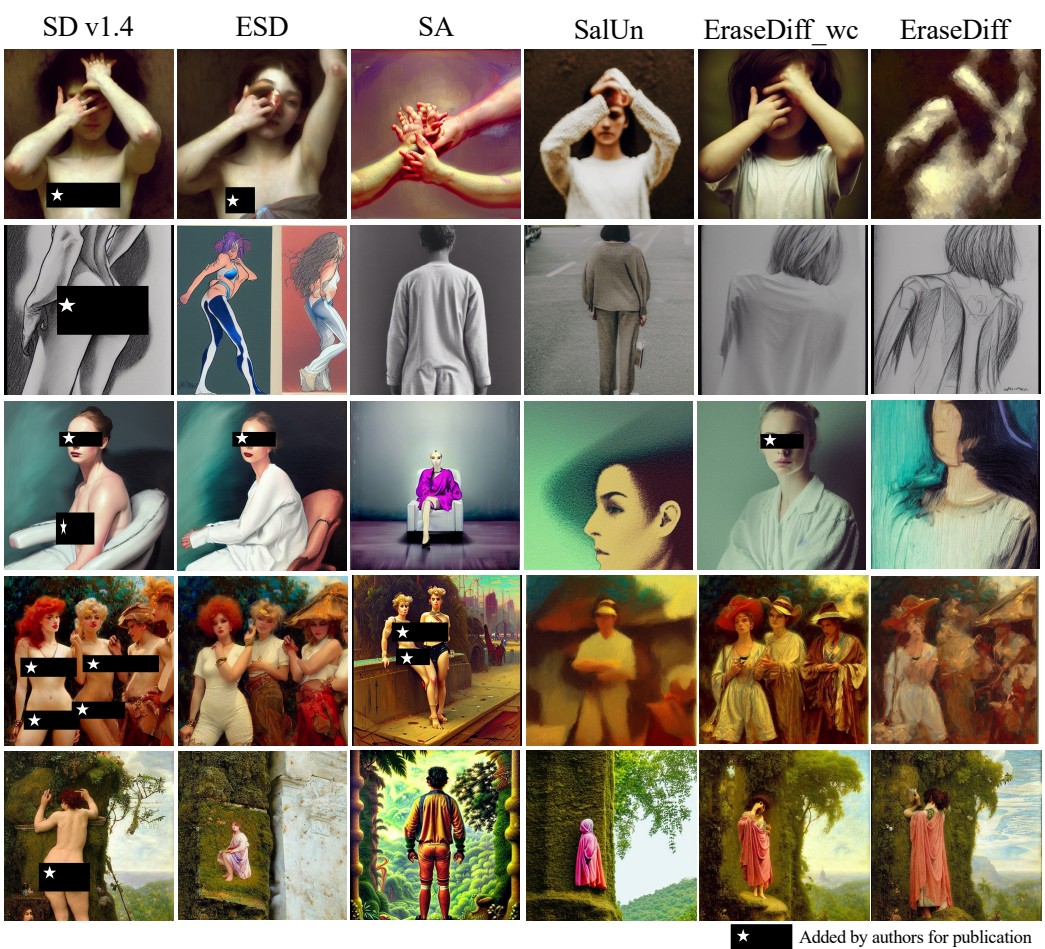

Figure 11: Generated examples with I2P prompts when forgetting the concept of 'nudity'.

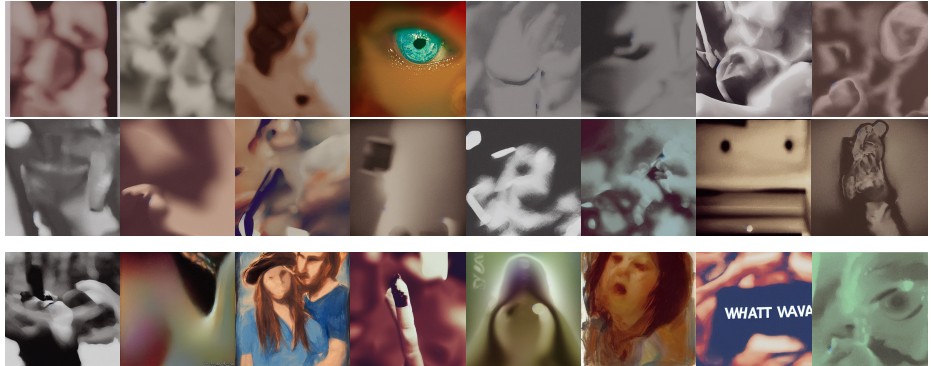

Figure 12: The flagged images generated by *EraseDiff* that are detected as exposed female breast/genitalia by the NudeNet classifier with a threshold of 0.6. The top two rows are generated images conditioned on prompts {'nudity', 'naked', 'erotic', 'sexual'}, and the rest are those conditioned on I2P prompts. No images contain explicit nudity content.

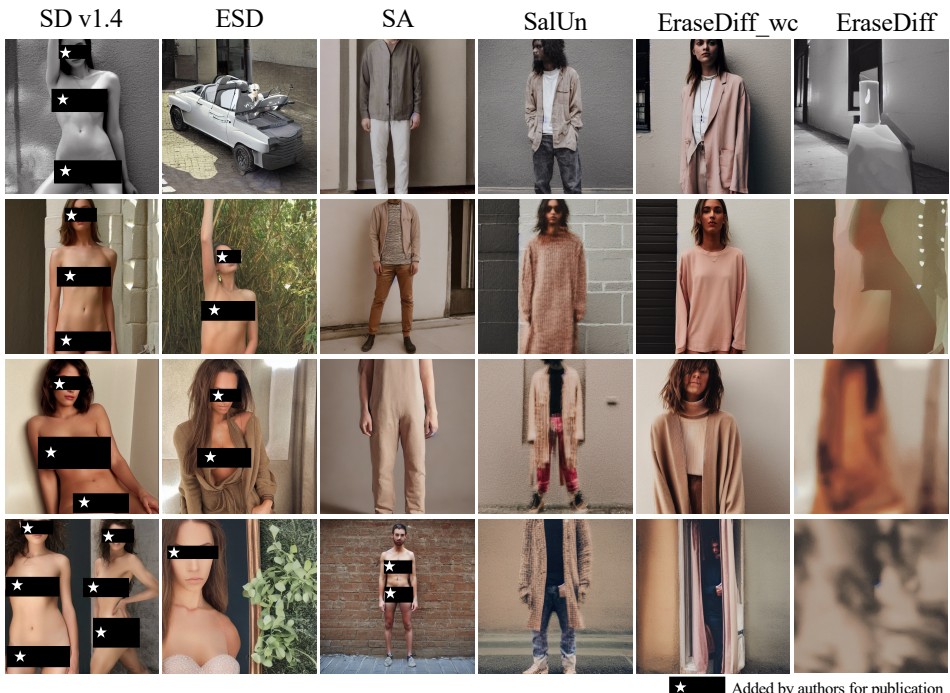

Figure 13: Visualization of generated examples with prompts {'nudity', 'naked', 'erotic', 'sexual'} when forgetting the concept of 'nudity'.

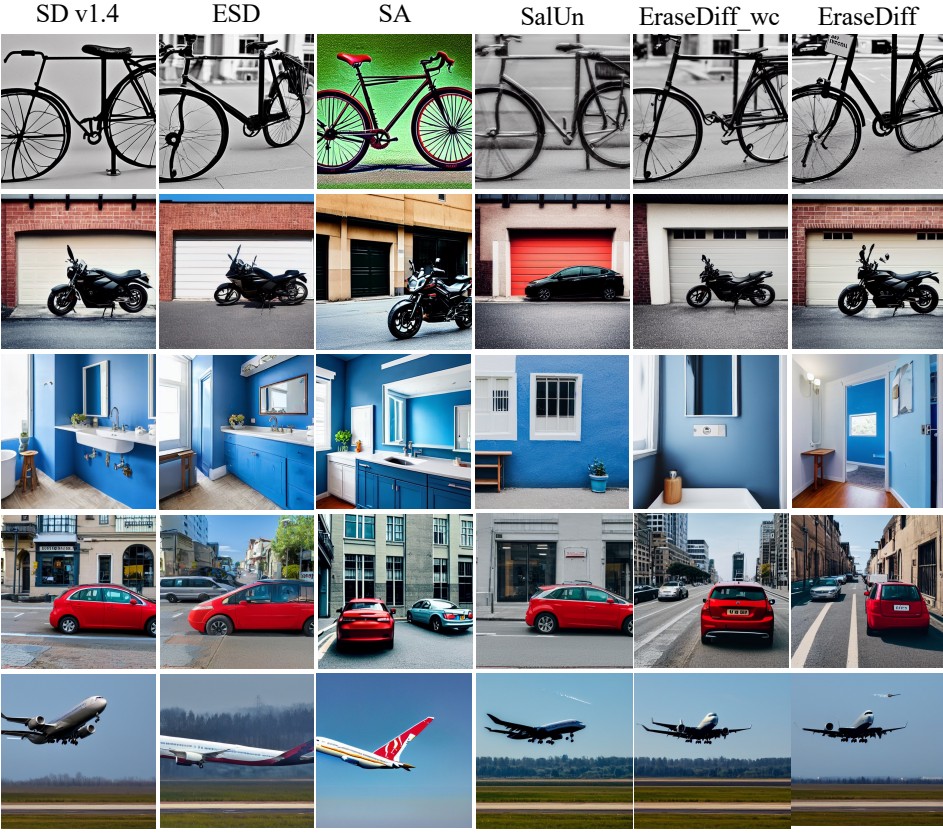

Figure 14: Visualization of generated images with COCO 30K prompts by the scrubbed SD models when forgetting the concept of 'nudity'.

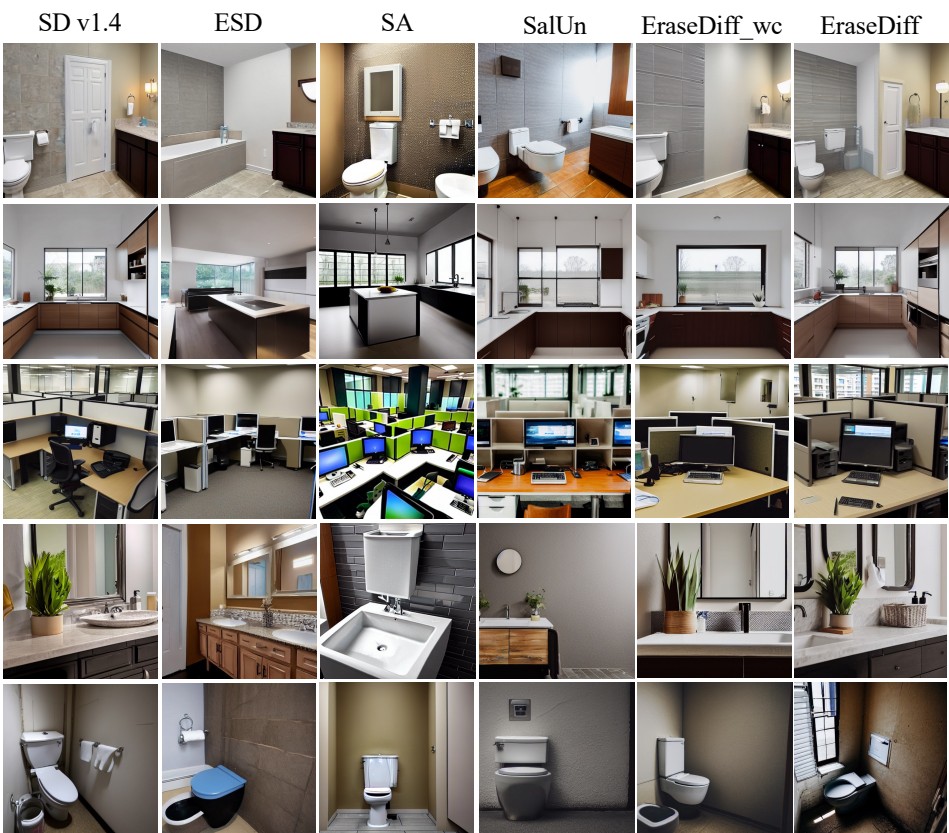

Figure 15: Visualization of generated images with COCO 30K prompts by the scrubbed SD models when forgetting the concept of 'nudity'.

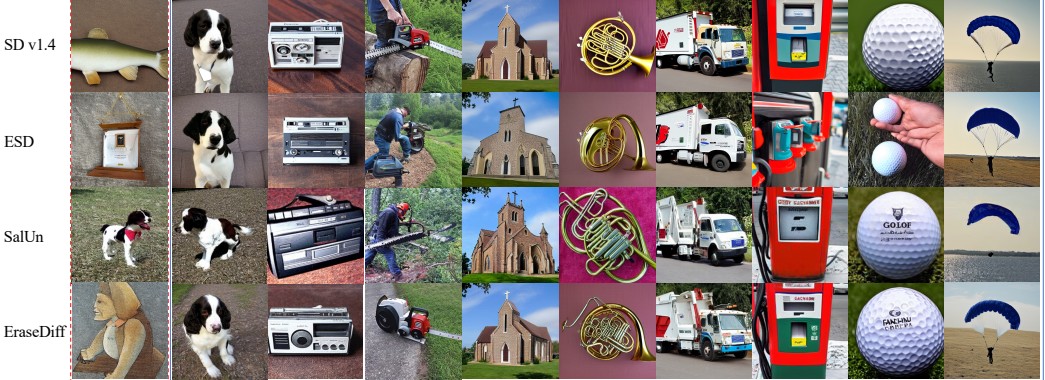

Figure 16: Visualization of generated images by the scrubbed SD models when forgetting the class 'tench' on Imagenette. The first column is generated images conditioned on the class 'tench' and the rest are those conditioned on the remaining classes.

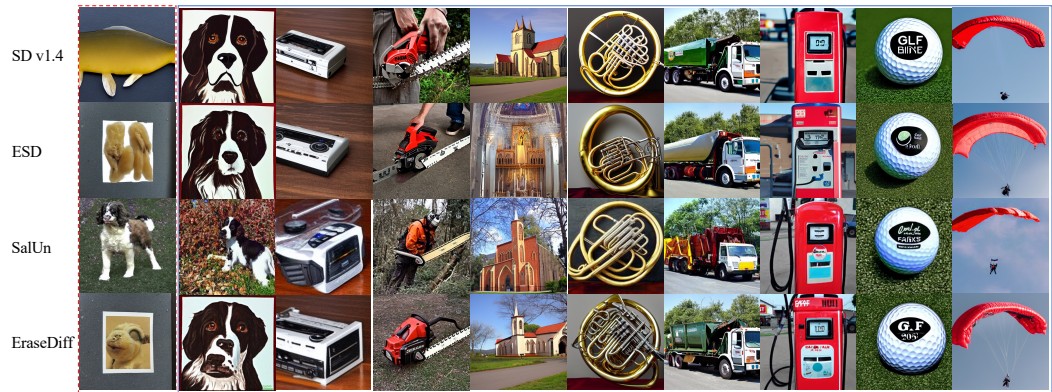

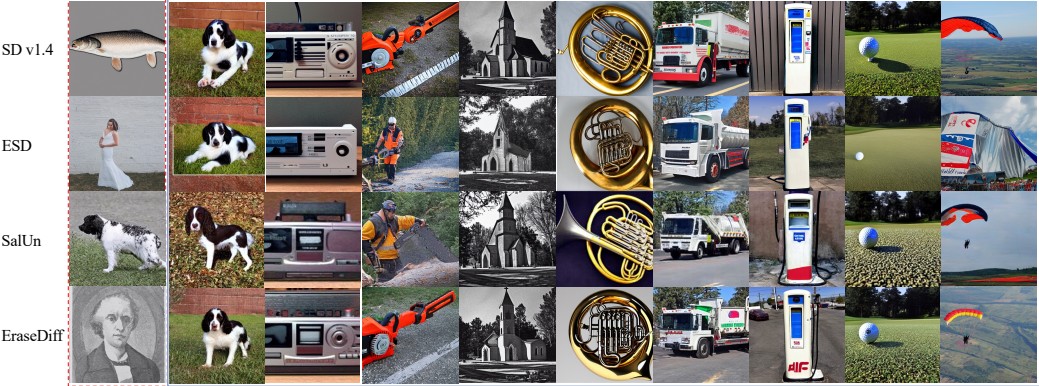

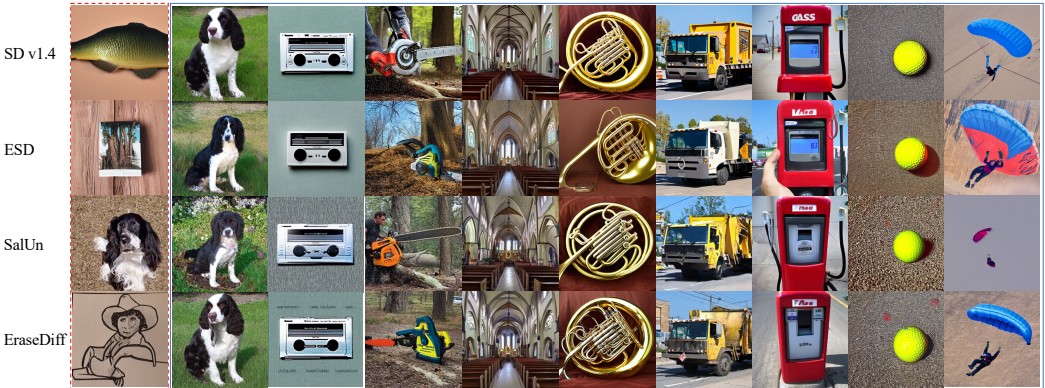

Figure 17: Visualization of generated images by the scrubbed SD models when forgetting the class 'tench' on Imagenette. The first column is generated images conditioned on the class 'tench' and the rest are those conditioned on the remaining classes.

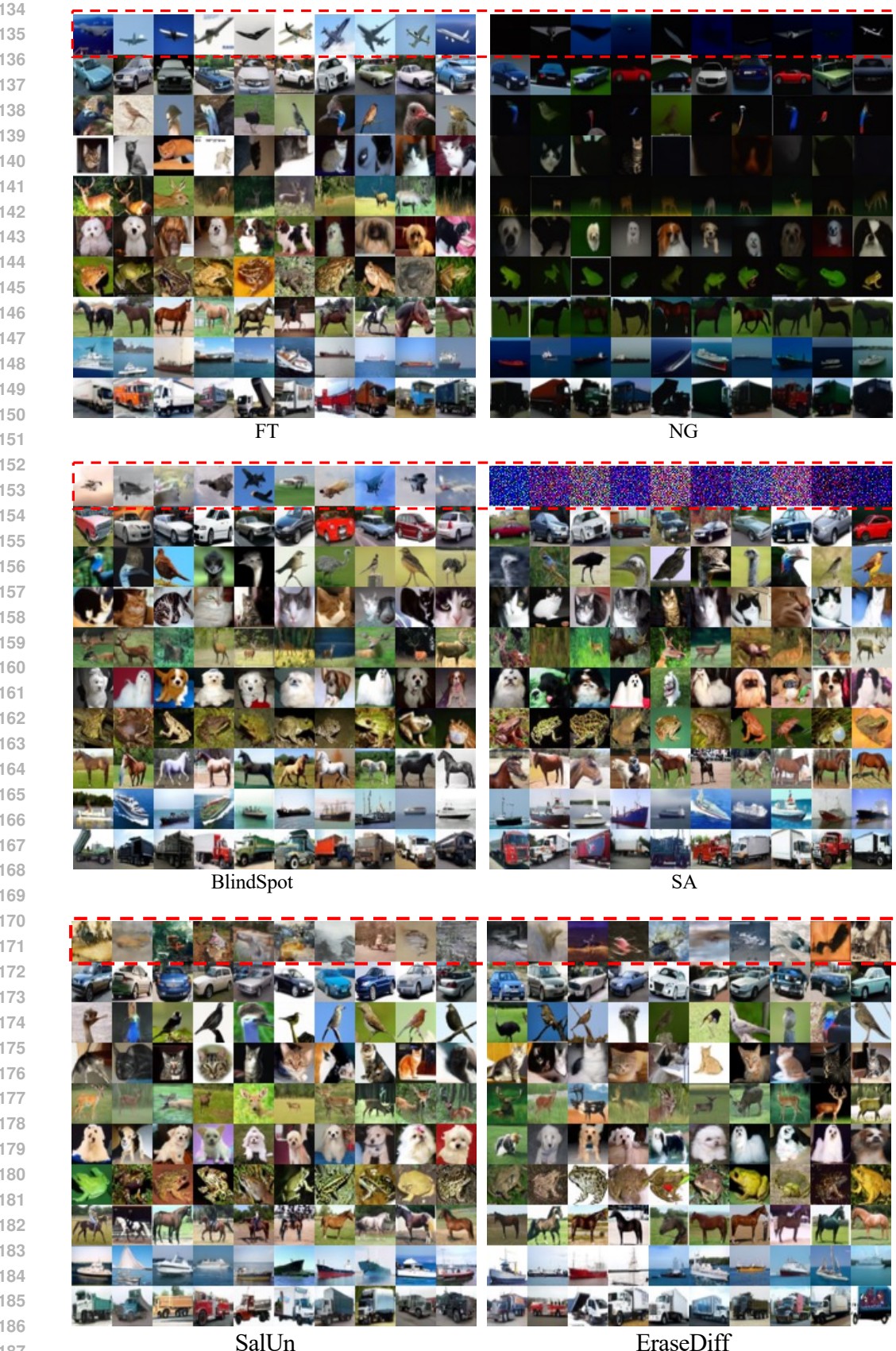

Figure 18: Visualization of generated examples when forgetting the class 'airplane' on DDPM.

