# OpenReview forum: "EraseDiff: Erasing Data Influence in Diffusion Models"
_ICLR.cc/2025/Conference — ICLR 2025 Conference Withdrawn Submission_

### Official Review · Reviewer_CQzs · 2024-10-19

**Soundness:** 2
**Presentation:** 2
**Contribution:** 2
**Rating:** 3
**Confidence:** 2

**Summary:**

This paper introduces EraseDiff, an unlearning method for diffusion models. The paper formulates the unlearning problem using a constraint optimization problem, which is approximated by a first-order method to solve. The paper compares EraseDiff with other methods using different images.

**Strengths:**

- The problem of unlearning for privacy and copyright considerations is significant.
- Source code is provided.
- The proposed EraseDiff method seems to be computationally friendly.

**Weaknesses:**

- No standard deviation is reported in the result tables.
- The efficacy of the method empirically is not convincing. For example, using 99.91 to claim a win over 99.88 is not convincing in Table 3.
- The loss function is not convincing. In the last paragraph of the introduction, the authors claim "minimizing the loss over the remaining data while maximizing that over the forgetting data". However, Eq. (2) is very similar to Eq. (1), and is still minimized in Eq. (3).
- Line 200 states "It is well known that" but a reference is still needed, missing here.
- The concept "unlearning" is not clearly defined, especially in the introduction part.
- In Eq.(6), $a_t$ is not explained, there should be at least a sentence like "for some fixed value $a_t$".
- Eq. (6) is not well motivated or explained.
- It is unclear from Table 1 that EraseDiff leads the performance.
- (minor) The authors do not need to submit a separate supplementary file as the appendix is already included in the main submission.

**Questions:**

- Why is Eq. (2) of this form? Should it be maximized instead?
- What is the performance on other diffusion models other than the few ones listed?
- (bonus) What is the relationship between solving for Eq. (5) and bilevel optimization?

---

### Official Review · Reviewer_Fvw2 · 2024-10-24

**Soundness:** 2
**Presentation:** 1
**Contribution:** 1
**Rating:** 3
**Confidence:** 3

**Summary:**

The authors study the problem of machine unlearning problem for diffusion models. They propose an unlearning approach that exhibits better computational efficiency than prior works.

**Strengths:**

- The problem of machine unlearning for diffusion models is important.

**Weaknesses:**

- The clarity of the paper should be greatly improved.
- The problem is not well and clearly defined until the experiment section.
- Methodology-wise the contribution to prior works seems limited.

## Detail comments
While I agree that the authors study a very important and timely problem on machine unlearning for diffusion models, I found that the contribution and quality of the paper do not meet the bar of ICLR. Firstly, I find the clarity of the paper in general should be greatly improved, especially on the rigor of the notations. For instance, what exactly is $\epsilon_f$ in equation (2)? It is never rigorously defined in the paper. Why $L_f$ in equation (4) can take an additional undefined argument $\phi_{init}$ compared to equation (2)? Why the expectation in equation (2) depends on $\epsilon$ when it does not appear anywhere else in equation (2)? What exactly is $\epsilon_\theta$ and what does the author mean by $\epsilon_\theta(x,t), \epsilon_\theta(x|c)$ and why is there multiple definitions of it? Note that I roughly get what the authors try to say but that is only because I am familiar with diffusion models. I feel the author should at least be rigorous in the definition of these basic terms as they are crucial for understanding the proposed method.

Another important issue is that the problem that the authors try to solve is never clearly well-defined. It is only clear to me until the experiment section that the authors want to modify the model so that it does not generate images pertaining to some labels or concepts. However, the way the authors introduce their method makes me feel that they aim to remove the influence of $D_f$ defined by certain labels or concepts to the model. Note that these two problems are very different, and I feel the authors do not convey clearly which goal they are trying to achieve. This also dulls the intuition and the reason why the proposed method makes sense in the first place.

In summary, I feel the paper need at least a major revision and I hope the authors can take time to polish their paper.

**Questions:**

1.	Why the expectation in eq 2 also depends on $\epsilon$? It does not have it anywhere.
2.	What is $\phi_{init}$? Why can $L_f$ in equation (4) take one more undefined argument than the one defined in (2)?
3.	What exactly is the underlying “unlearning” problem? Do the authors aim to erase some concepts? If so, what is the problem formulation?
4.	What exactly is the design of $\epsilon_f$?

---

### Official Review · Reviewer_g9PU · 2024-11-03

**Soundness:** 2
**Presentation:** 3
**Contribution:** 2
**Rating:** 5
**Confidence:** 2

**Summary:**

The paper addresses the problem of unlearning specific data influences in
generative models (here, diffusion models) to mitigate privacy risks
associated with data memorization. EraseDiff, the proposed methods, frames
unlearning as a constrained multi-objective optimization problem that
allows the model to retain its performance on preserved data while
selectively removing the influence of data flagged for deletion. This
approach involves adjusting the model’s generative process to diverge from
the typical denoising procedure when encountering the data to be forgotten
by choosing a distribution different from the standard normal distribution
used for the rest of the dataset.  A first-order optimization method is
proposed to address the computational complexity inherent in diffusion
processes. Extensive experiments and comparisons with existing algorithms
indicate that EraseDiff maintains the model’s utility while achieving
effective and efficient data unlearning.

**Strengths:**

Pros:
- Well-motivated problem
- Clear problem formulation
- Rigorous theoretical approach with pareto-optimal guarantees.
- Comprehensive review of the literature relative to proposed method.

**Weaknesses:**

Cons:
- Could benefit from additional experimentation in some aspects.
- Some parts need to be clearly explained.
- Performance not very different from one of the baselines.

The training objective part is not very clear. Equation (1) is fine. But
equation (2) is says epsilon is sampled from normal distribution, but later
in the equation, epsilon_f is mentioned. What is the relationship of
epsilon_f with epsilon? Subsequently, it is also mentioned that epsilon_f
is chosen to be a different from epsilon. What does "This could be ..."
sentence mean? What was really used to confound the approximator? Does
equation (4) correspond to a local minima?

Table 1 does not really indicate that the EraseDiff is much better than the
baselines. Also, the authors only chose one class (airplane) for this line
of experimentation. They could have considered more instances to show a
more comprehensive ealuation.

Figure 3 and Table 2 indicate that SalUn performs very close to Erasediff,
if not better in some aspects.

**Questions:**

Questions inserted in "Weaknesses".

---

### Official Review · Reviewer_YNgS · 2024-11-06

**Soundness:** 2
**Presentation:** 3
**Contribution:** 2
**Rating:** 5
**Confidence:** 3

**Summary:**

This work proposes an algorithm for unlearning in diffusion models. Unlike prior work, which formulate the optimization problem as minimizing a sum of two losses - one for the remember set and one for the forget set - this work proposes a bi-level optimization problem. They derive the parameter update rule for this optimization problem. Experiments are performed on three tasks which demonstrate class and concept wise forgetting with mixed results.

**Strengths:**

1) The problem is important and relevant to machine unlearning.

2) The proposed method is novel in its approach.

3) Experiments consider interesting, relevant tasks, and the proposed method demonstrates a reduction in computation time compared to competitive baselines.

4) Figure 2 illustrates EraseDiff empirically reduces gradient conflict on the CIFAR 10 dataset.

**Weaknesses:**

See questions.

**Questions:**

1) Line 205: It is not clear what this expression means. Does it mean \( \nabla_{\phi = \theta} =  L(\phi, D_f) = 0 \)? Or is it for some \(\phi\) which can be reached by an optimization algorithm after initializing the parameters at \(\theta\)? In that case, would that depend on the optimization algorithm used, number of steps, random seed etc?

2) Related to above, what is the theoretical justification for formulating this optimization problem? How does it relate to traditional definitions of unlearning?

3) For the experiments (in Table 1 and 2), were they run over multiple random seeds? If yes, could the error bars and standard deviations be reported? Without those, it is hard to judge the significance of the results. For example, Line 399 mentions ‘there is a decrease in recall (diversity)’ when comparing EraseDiff to SA. However, error bars would be required to judge the scientific significance of this statement.

4) Related to above, the experiments do not seem to show improvement in quality of generated images or ability to forget compared to some baselines. For Table 1, SA outperforms EraseDiff in both FID and \( P_{\psi}(y = c_f | x_f) \) while they are nearly equal in precision and recall. For Table 2, ESD has a lower FID and nearly equal CLIP score.
Minor comments (did not affect rating):

5) Line 234: For clarity and completeness, showing the steps of how Eq 5 can be formulated as Eq 6 (using Liu et al.) might be better. This can be done in the appendix if space is a constraint.

---

### Note · Authors · 2024-11-13

I have read and agree with the venue's withdrawal policy on behalf of myself and my co-authors.